

# Expanding the utility of the dextran sulfate sodium (DSS) mouse model to induce a clinically relevant loss of intestinal barrier function

Kyle E. Cochran[1], Nicholas G. Lamson[1] and Kathryn A. Whitehead[1,2]

[1] Department of Chemical Engineering, Carnegie Mellon University, Pittsburgh, PA, United States of America
[2] Department of Biomedical Engineering, Carnegie Mellon University, Pittsburgh, PA, United States of America

## ABSTRACT

**Background**. Inflammatory bowel disease (IBD) is a family of debilitating disorders that affects more than 1 million people in the United States. Many animal studies of IBD use a dextran sulfate sodium (DSS) mouse model of colitis that induces rapid and severe colitis symptoms. Although the typical seven-day DSS model is appropriate for many studies, it destroys intestinal barrier function and results in intestinal permeability that is substantially higher than what is typically observed in patients. As such, therapies that enhance or restore barrier integrity are difficult or impossible to evaluate.
**Methods**. We identify administration conditions that result in more physiologically relevant intestinal damage by systematically varying the duration of DSS administration. We administered 3.0% DSS for four to seven days and assessed disease metrics including weight, fecal consistency, intestinal permeability, spleen weight, and colon length. Histology was performed to assess the structural integrity of the intestinal epithelium.
**Results**. Extended exposure (seven days) to DSS resulted in substantial, unrecoverable loss of intestinal structure and intestinal permeability increases of greater than 600-fold. Attenuated DSS administration durations (four days) produced less severe symptoms by all metrics. Intestinal permeability increased only 8-fold compared to healthy mice, better recapitulating the 2–18 fold increases in permeability observed in patients. The attenuated model retains the hallmark properties of colitis against which to compare therapeutic candidates. Our results demonstrate that an attenuated DSS colitis model obtains clinically relevant increases in intestinal permeability, enabling the effective evaluation of therapeutic candidates that promote barrier function.

## INTRODUCTION

Inflammatory bowel disease (IBD) is a set of painful and debilitating disorders that affects approximately 1.6 million Americans, with upward of 70,000 new diagnoses each year (*Loftus, 2004*; *Kaplan, 2015*; *Colombel & Mahadevan, 2017*). IBD includes ulcerative colitis, Crohn's disease, and indeterminant colitis, and our understanding of the etiology of these diseases is lacking. Patients with IBD suffer from persistent diarrhea, rectal bleeding,

Corresponding author
Kathryn A. Whitehead,
kawhite@cmu.edu

weight loss, fatigue, and abdominal pain, which dramatically impacts their health and quality of life (*Strober, Fuss & Mannon, 2007*). Current treatments manage, rather than cure, IBD and include antibiotics, steroids, and immunosuppressants (*Pithadia & Jain, 2011*; *Triantafillidis, Merikas & Georgopoulos, 2011*). Although the full mechanisms through which the disease develops remain uncertain, evidence points to loss of intestinal barrier function as a key aspect of clinical IBD. It is unclear if barrier loss is a symptom or a cause of the disease (*Martini et al., 2017*; *Lee et al., 2018*). Thus, reducing intestinal permeability has been the goal of some therapeutic candidates (*Arrieta et al., 2009*), and the success of such studies is dependent on the use of appropriate animal models.

The most widely used IBD model uses dextran sulfate sodium (DSS) that is introduced into the drinking water of mice to chemically induce colitis. Although its exact mechanism for inducing colitis is unknown, DSS administered in the drinking water of mice inflicts damage to the intestinal epithelium (*Chassaing et al., 2014*; *Kiesler, Fuss & Strober, 2001*). This results in colitis symptoms including loose and bloody stool, significant weight loss, shortening of the colon, and decreased intestinal barrier function (*Chassaing et al., 2014*; *Eichele & Kharbanda, 2017*; *Mallon et al., 2010*). This model is the current standard in many laboratories because it is relatively reproducible, customizable, and doesn't require the spontaneous development of colitis that models using transgenic or knockout mice do (*Eichele & Kharbanda, 2017*). However, a significant downfall of the model is the rapid and widespread damage to the intestinal epithelium, which doesn't reflect the degree of decreased barrier function in patients (*Kiesler, Fuss & Strober, 2001*).

Studies of potential IBD therapeutics often use 3.0–5.0% DSS for longer durations of six to eight days (*Zhang et al., 2015*; *He et al., 2016*). While this falls within the recommendations of administration protocols (*Chassaing et al., 2014*), the rapid and severe colitis and epithelial damage induced by these models does not resemble clinical IBD (*Kiesler, Fuss & Strober, 2001*). Specifically, these models result in loss of intestinal barrier function that is far more severe than the losses observed in clinic. Lactulose/mannitol tests have demonstrated intestinal permeability increases of 2- to 18-fold in colitis patients, while the standard DSS model can increase permeability to macromolecule markers by more than 100-fold (*Welcker et al., 2004*; *May, Sutherland & Meddings, 1993*). Studies of therapeutics that restore intestinal barrier function have been limited, likely in part because the standard DSS model destroys the intestinal epithelial monolayer. Fortunately, previous research suggests that it is possible to induce milder colitis symptoms by reducing DSS administration duration (*Yan et al., 2009*). We were motivated to expand upon these findings by evaluating a broad array of colitis metrics, including intestinal permeability, in an attenuated DSS mouse model.

To identify DSS colitis protocol conditions that better recapitulate the loss of barrier function present in clinical IBD, we fixed the DSS dose at 3.0% and varied exposure length from four to seven days. We tracked the onset and progression of weight loss, fecal samples, colon length, spleen weight, and intestinal permeability and scored colon histological samples. In addition to identifying DSS administration conditions that result in clinically relevant levels of intestinal permeability, we also examined how environmental factors, such as acclimation to experimental conditions, affected colitis onset.

## MATERIAL AND METHODS

### Animals

Animal protocols were approved by the Institutional Animal Care and Use Committee at Carnegie Mellon University (Pittsburgh, PA), and all experiments were conducted in accordance with approved protocol AR201900009. Furthermore, all studies were conducted in accordance with local, institute, and federal regulations for vertebrate animal research. Female C57BL/6 mice six weeks of age were purchased from Charles River Laboratories and acclimated to facility conditions for two weeks before studies. Mice were housed under controlled temperature (25 °C) on 12 h light-dark cycles. Animals were given access to standard diet and water, with experimental groups receiving 3% DSS in their water. Animals were euthanized early if they lost more than 30% of their initial body weight or if they showed severe signs of distress. All animals were euthanized at the end of the study by $CO_2$ inhalation.

### Materials

Dextran sodium sulfate colitis grade (DSS, 36,000–50,000 MW) was purchased from MP Biomedicals (Santa Ana, CA). Fluorescein isothiocyanate-dextran 4 kDa (FITC-DX4) was purchased from Sigma Aldrich (St. Louis, MO). Hemoccult Guaiac Fecal Occult Blood Test slides were purchased from VWR (Radnor, PA). Phosphate buffered saline (PBS) was purchased from Thermo Fisher LifeTech (Carlsbad, CA). Water bottles, spouts, cages, and bedding were provided by Mellon Institute Centralized Vivarium.

### DSS administration

Glass water bottles with stainless steel drinking spouts were filled with 100 mL of chlorine treated water and placed in each group's cage. For groups receiving DSS, 3.0 grams of DSS was fully dissolved in the water before the bottle was placed the cage. Water level for each cage was recorded daily and the water was fully changed every two days. Animals received DSS for 7, 6, 5, or 4 days, after which they were switched to fresh water for 7 days unless early euthanasia was required. Groups were designed so that DSS administration was staggered and all animals were switched to fresh water on the same day.

### Weight and fecal scores

Weight was recorded and a small fecal sample was collected for each animal daily. Fecal samples were examined for consistency, tested for blood using Hemoccult slides, and scored accordingly.

### Intestinal permeability

Mice were rectally administered FITC-DX4 as a colonic permeation marker (60 mg/mL in PBS, 600 mg/kg animal). Three hours later, blood samples were taken via the submandibular gland and the serum was examined for fluorescence at 495/519 nm on a BioTek Synergy2 plate reader. Application of a calibration curve yields a blood concentration of FITC-DX4.

### Evaluation of colons and spleens

Upon euthanasia, the spleen and colon of each animal were excised. Colons were measured to the nearest half millimeter then fixed in 4% formaldehyde for 24 h. Fixed colons were

washed twice with PBS and subsequently stored in 70% ethanol at 25 °C. Hematoxylin and eosin (H & E) staining was performed on colon cross-sections by University of Pittsburgh Medical Center Tissue and Research Pathology Services (Pittsburgh, PA). Stained slides were scored and imaged by Dr. Lora Rigatti, University of Pittsburgh (Pittsburgh, PA). Spleens were weighed on an Ohause Adventure Pro analytical balance.

## Statistical analysis

Animals were placed in group of 6 per cage, and premature animal deaths were accounted for in analysis. Error is displayed in all plots as the standard error of the mean and experimental groups are compared to the control individually using ordinary $t$-test, without correction for multiple comparisons.

## RESULTS

The use of an animal model that recapitulates key aspects of colitis is crucial to the study of disease mechanisms and evaluation of potential IBD therapeutics. One of the most common colitis models, the DSS model, is highly tunable and can be used to induce colitis of varying severities. We reasoned, therefore, that it would be possible to induce colitis symptoms without destroying intestinal barrier function.

### Establishing a model with less severe symptoms

To identify a chemically-induced colitis protocol that results in less severe, yet measurable, symptoms, we examined how altering the administration duration of DSS affected symptom onset. To accomplish this, groups received 3.0% DSS in their drinking water for either seven, six, five, or four days. After the DSS administration period, they were switched to fresh water for seven days, or four days if early euthanasia was required. Starting with the seven-day group on Day 0, each group began receiving DSS on subsequent days, such that all animals were switched to fresh water on Day 7 (Fig. 1A). Each day, mice were weighed, and fecal samples were collected and scored. All DSS animals began losing substantial weight four to five days after the start of DSS administration (Fig. 1B), but only the four-day DSS group made any recovery before euthanasia was required. The seven, six, and five-day groups were unable recover body weight, and each required early euthanasia because they reached 30% weight loss (a predetermined endpoint of the study). Average weight loss per day was calculated between the fourth and eighth day of DSS administration. Average daily weight loss was similar between colitis groups, but animals from the four-day DSS group lost less weight per day than animals from the five-day DSS group ($p = 0.0068$, Fig. 1C).

Each day, fecal samples were taken and scored for consistency and blood content (Figs. 1D and 1E). Colitis is indicated by the increasing fecal scores observed for all DSS groups. As with weight loss, the onset of increasing fecal scores was similar across all colitis groups, but only the four-day DSS group recovered after returning to fresh water. Together, these results indicate that shortening the DSS administration time affected the severity, rather than the onset, of colitis symptoms such as weight and fecal score.
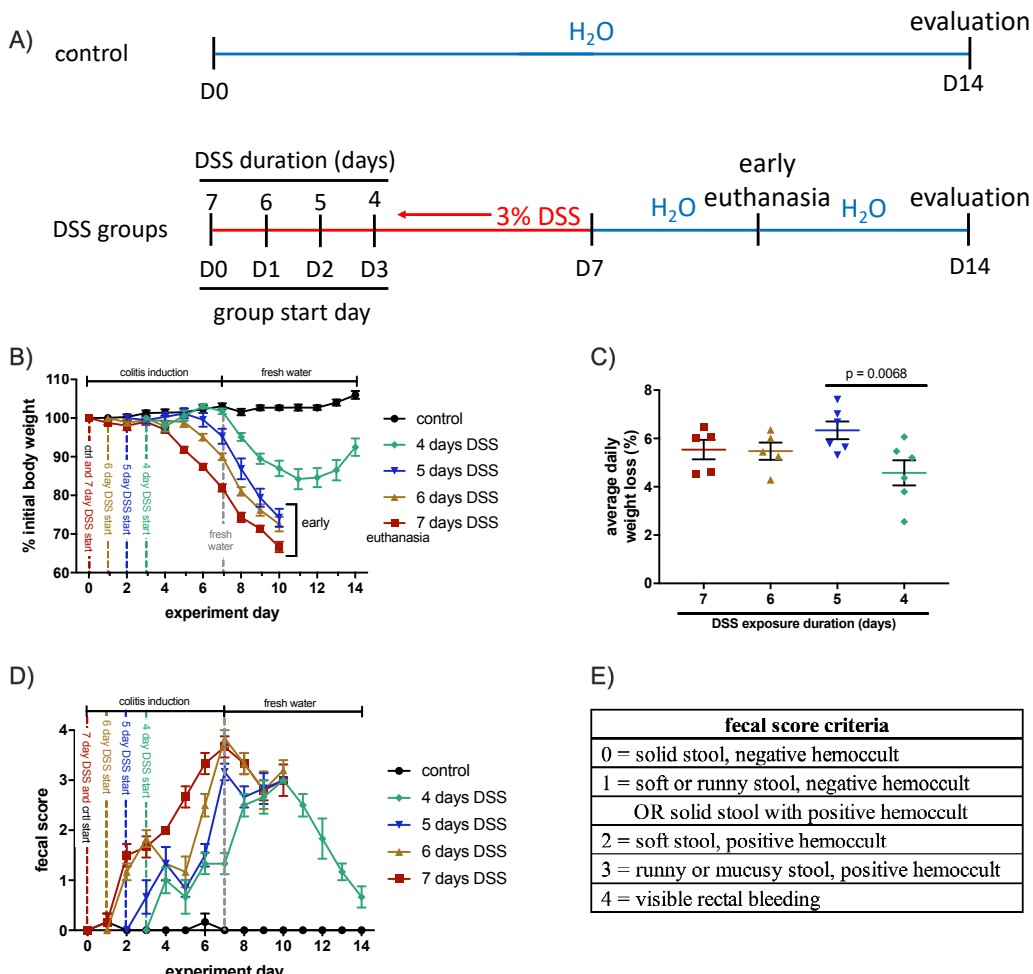

**Figure 1 Weight loss and fecal scores were affected by the duration of DSS exposure.** (A) DSS was administered to the seven-day DSS group on D0, followed by the seven-day group on D1, and so on. Early euthanasia was required for mice in the seven-, six-, and five-day DSS groups. (B) Mice from each group began losing weight four to five days after the start of their DSS regimen. Weight loss from all groups continued after switching to fresh water, with all but the four-day DSS group reaching the 30% weight loss cutoff. Only the four-day DSS group recovered body weight. (C) Average daily weight loss was calculated for each animal, between the fourth and eighth days of their DSS regimen. Weight loss was similar across groups, with the four-day group losing the least weight per day. (D) Fecal scores increased as the DSS administration duration lengthened. (E) Fecal samples were scored daily and ranged from 0–4, with 4 indicating the most severe symptoms. Error bars for all panels display s.e.m. ($n = 5$–6).

## Effect of DSS administration duration on intestinal permeability

Decreased intestinal barrier function is a symptom of clinical IBD and IBD animal models (*Chassaing et al., 2014*; *Michielan & Incà, 2015*). To demonstrate the loss of intestinal barrier function during colitis development, we rectally administered the permeability marker FITC-DX4 (4 kDa) on Days 7, 10, and 14 and collected blood samples three hours later. The concentration of this fluorescent permeability marker in serum samples was analyzed to assess the barrier function of the animals' colons.

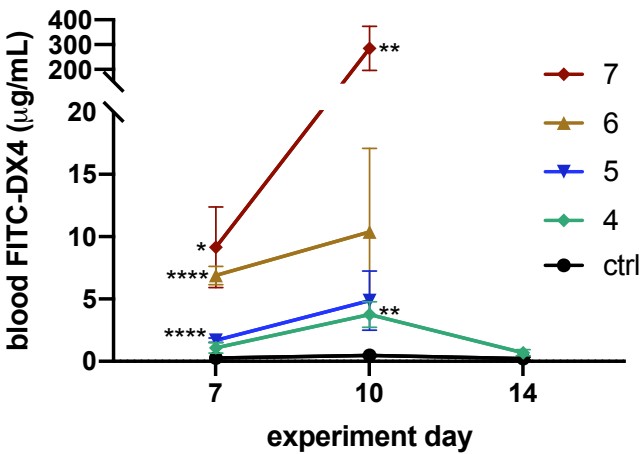

**Figure 2 DSS colitis increased intestinal permeability, the severity of which depended on DSS exposure duration.** Intestinal permeability measured at Day 7 was significantly increased for all groups except the four-day DSS group. By Day 10, the permeability of the seven-day DSS group was nearly 28-fold higher than any other DSS group. Seven days after being switched to fresh water (Day 14), the intestinal permeability of the four-day DSS group returned to normal. Mice that received DSS for only four days experienced a significant increase in permeability but recovered to baseline after seven days of fresh water. *$p < 0.05$, **$p < 0.01$, ****$p < 0.0001$ compared to the PBS control, error bars display s.e.m. ($n = 5$–6).

On Day 7, the day on which each group completed their DSS regimen and was switched to fresh water, the seven- ($p = 0.005$), six- ($p < 0.0001$), and five-day ($p < 0.0001$) DSS groups each had significantly elevated intestinal permeabilities (Fig. 2). The four-day DSS group had permeabilities that were slightly higher than the control, while the seven-day DSS group showed the most intense increase in permeability, increasing 35-fold compared to the control. Permeability measurements on Day 10, three days after animals were switched to fresh water, showed that barrier function continues to decline even when DSS is discontinued. At this point, the seven-day group showed a greater than 600-fold increase in intestinal permeability compared to the control ($p = 0.0063$), while the six ($p = 0.1364$), five ($p = 0.0936$), and four-day ($p = 0.0089$) DSS groups all had less than a 22-fold increase (Fig. 2).

Because animals from the seven-, six-, and five-day DSS groups reached greater than 30% body weight loss around Day 10, they required early euthanasia. Only the control and four-day DSS groups continued through the entire 14-day experiment. On Day 14, the final intestinal permeability measurement for the four-day DSS group showed that animals had returned to near control values, indicating a restoration of colonic barrier function (Fig. 2). Overall, the abbreviated four-day DSS protocol results in a measurable increase in intestinal permeability without permanently destroying intestinal barrier function.

## Effect of DSS administration duration on spleen weight and colon length

As colitis progresses and the epithelium begins to erode, the colon thins and becomes shorter and immune cells infiltrate the lamina propria (*Chassaing et al., 2014*). Infiltration of immune cells results in inflammation that is often assessed by measuring spleen weight.

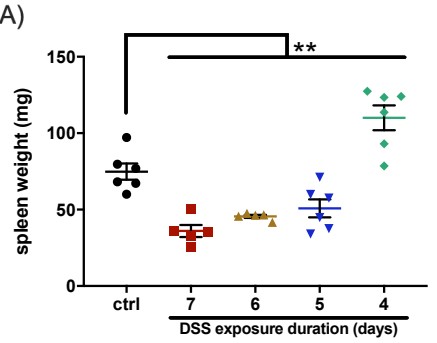
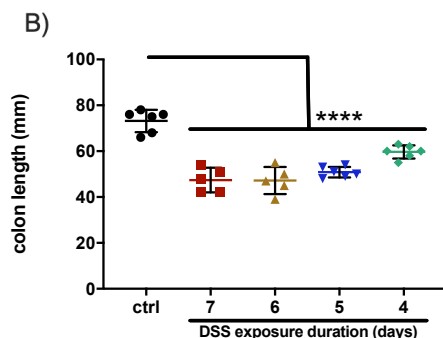

**Figure 3** **Only animals from the four-day DSS group experienced an increase in spleen weight, but all DSS groups had significant shortening of the colon.** (A) The spleens from all other groups weighed significantly less than the control group, likely due to the animals' severe dehydration and a lack of recovery from colitis. (B) Although a four-day DSS regimen resulted in less severe symptoms by most metrics, it nonetheless produced measurable colitis. For both panels, $*p < 0.05$, $****p < 0.0001$, error bars display s.e.m. ($n = 5$–6).

These phenomena provide two additional metrics, colon length and spleen weight, by which the severity of chemically-induced colitis can be measured (*Chassaing et al., 2014*). To examine these aspects of the disease model, mice were euthanized on the final day of the experiment and their colons and spleens were harvested. Spleen weights from all groups differed significantly from the control ($p = 0.0003$ for seven-day, $p = 0.0008$ for six-day, $p = 0.0127$ for five-day, $p = 0.0047$ for four-day), but only the four-day DSS group experienced enlarged spleens (Fig. 3A). The seven-, six-, and five-day DSS groups each had spleen weights that were significantly less than the control. Dehydration and lack of a recovery (requiring early euthanasia) are likely the reasons that groups receiving DSS for longer durations had lower spleen weights.

The colons of animals from all DSS groups were significantly shorter than control animals ($p < 0.0001$ for all groups, Fig. 3B). Although the symptoms of the four-day DSS protocol were less severe, thinning and shortening of the colon nonetheless occurs. This indicates that colonic shortening, one of the hallmarks of chemically induced colitis, is retained in this attenuated four-day model and provides a viable metric against which to compare IBD therapeutics.

## Histological evaluation of colon samples from the colitis model

To assess the effect that DSS administration duration has on the degree of damage to the intestinal epithelium, three colon samples from each group were stained with hematoxylin and eosin (H & E) and scored. Each sample was blindly scored by a trained pathologist for four criteria shown in Fig. 4A: inflammation (0-3), extent (0-3), regeneration (3-0), and crypt damage (0-4). The score from each criterion was multiplied by a score for the involvement (0-3) to indicate how much of the sample section was affected. Possible scores ranged from 0–39, with higher scores indicating increased damage to the intestinal epithelium.

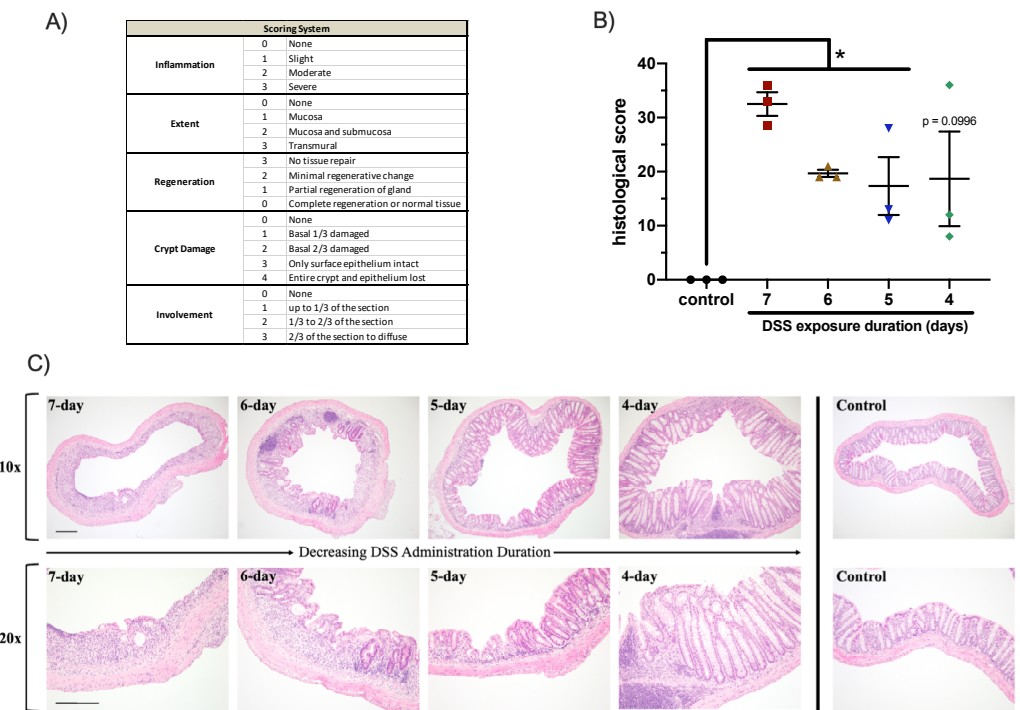

**Figure 4  DSS administration duration dramatically affected the integrity of the colonic epithelium.**
(A) Colon samples were evaluated for inflammation, extent, regeneration, crypt damage, and involvement and scored 0–39. Increasing scores indicate increasing damage inflicted from DSS administration.
(B) Epithelial and crypt structure were absent from the colon samples of animals exposed to 7 days of DSS. Colonic structure gained integrity as the duration of DSS exposure was reduced. $*p < 0.05$, error bars display s.e.m. ($n = 3$). (C) Colon samples were H & E stained and representative images were captured at 10× and 20× magnification (scale bars represent 100 μm).

Control mice had an average score of 0, indicative of the fully intact epithelium found in healthy mice (Fig. 4B). Mice that received DSS had individual scores ranging from 8 to 36, and scores generally increased with increasing length of DSS administration. The seven-day DSS group had the highest score with an average of 32.5 ($p < 0.0001$), followed by the six-day DSS group with an average of 19.7 ($p < 0.0001$). The five- and four-day DSS groups had the lowest average histological scores, 17.3 ($p = 0.0319$) and 18.7 ($p = 0.0996$), respectively. Images of selected colons in Fig. 4C show the increasing loss of epithelial structure as DSS administration duration increased. In the seven- and six-day DSS groups, lack of crypts further confirms the severity of the colitis associated with longer DSS administration regimens. Similar to other metrics, the histological scores show that the full seven-day DSS model resulted in substantial damage, unrecoverable to the intestinal epithelium. Reducing DSS administration time decreased the severity of the inflicted damage, although there was some variability within groups.

## Avoiding pitfalls: acclimation to experimental conditions

Chemically-induced colitis affects each animal differently, which manifests as highly variable results in these models. To reduce variability, animals should be given ample time

to acclimate to experimental conditions before observations begin. In our animal facility, DSS administration required removal of the standard Lixit drinking system, which was replaced with individual water bottles in each cage. Animals must be properly acclimated to this switch before experimental observations begin.

To demonstrate this, we set up control and DSS cages that were switched from the Lixit system to water bottles five days before observations (Day -5) or on the day that observation began (Day 0) (Figs. 5A and 5B). For control animals that were switched to water bottles on Day 0 (Fig. 5A), there was a significant drop off in weight of about 3% on Day 1 ($p = 0.0080$), likely due to hesitation by the mice to drink water from an unfamiliar system. In general, the weight of these mice was less consistent than that of the mice that were acclimated to water bottles five days before observations began. A similar pattern was observed for mice receiving DSS. On Day 1, mice that were not acclimated to the water bottle lost an average of 5% of their body weight (blue squares, Fig. 5B). Although these mice did recover that weight, their initial lack of water consumption resulted in a measurable delay in the onset of colitis related weight loss compared to acclimated mice (green circles). On each day after Day 4, acclimated mice had significantly lower weights compared to the control, ($p = 0.0060$ for Day 4, $p = 0.0065$ for Day 5, $p = 0.0288$ for Day 6, and $p = 0.0129$ for Day 7) indicating the faster and more consistent colitis onset. For this reason, we recommend that mice be acclimated to experimental conditions, including drinking from water bottles, at least five days prior to the experiment to allow them ample time to adjust.

## DISCUSSION

As IBD becomes increasingly prevalent and ever-growing efforts are devoted to understanding and treating the disease, it is imperative to fully understand IBD models and how to best implement them. DSS is often administered at high doses for longer durations to induce an acute colitis (*Zhang et al., 2015*; *He et al., 2016*). Our results confirm that this induces a sweeping loss of intestinal barrier function, which precludes the testing of therapeutics designed to repair the intestinal epithelium.

Decreased intestinal barrier function is a critical factor in clinical IBD and the DSS colitis model due to the subsequent immune cell infiltration and inflammatory response (*Eichele & Kharbanda, 2017*; *Michielan & Incà, 2015*). However, DSS administration inflicts widespread damage to the intestinal epithelium that results in permeability increases far higher than those observed in clinical IBD (*Welcker et al., 2004*; *May, Sutherland & Meddings, 1993*). Permeability increased in all groups receiving DSS, but the magnitude of the increase was predicated on administration duration. The four-day DSS protocol produced intestinal permeability changes that best mimic what is observed clinically. First, the 8-fold increase in intestinal permeability for the four-day group best recapitulates the 2- to 18-fold increases observed in IBD patients (*Welcker et al., 2004*; *May, Sutherland & Meddings, 1993*). Furthermore, permeability in this group returned to baseline before euthanasia. These results are more representative of many cases of clinical IBD where a patient experiences bouts of inflammation from which they can recover. Although

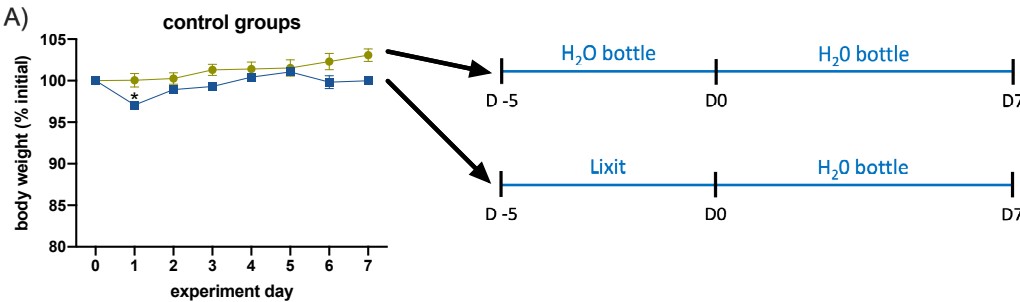

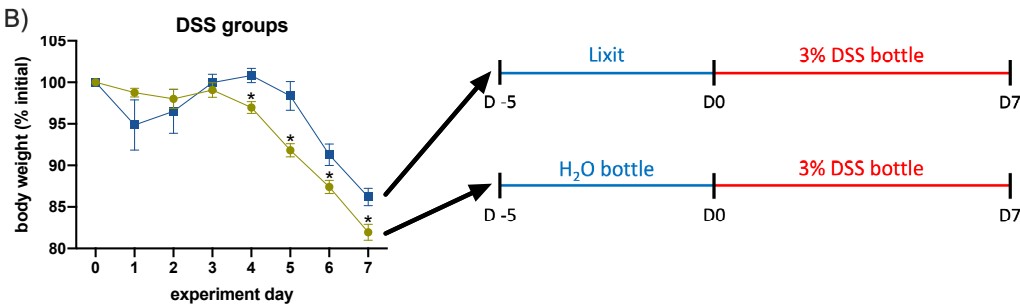

**Figure 5** **Failing to acclimate animals to experimental conditions resulted in significant differences in symptom onset.** (A) When not acclimated to water bottles, control mice (receiving only water) lost up to 3% of their body weight when they were switched from Lixit to bottle water consumption. (B) DSS-treated mice not acclimated to water bottles experienced greater fluctuations in body weight and delayed onset of symptoms compared to acclimated mice. For all panels, $n = 5$–$6$, with error bars representing s.e.m., $^* p < 0.05$.

the five-day group experienced similar increases in permeability (10-fold compared to control animals), their body weight loss required early euthanasia. In experiments in which epithelial treatments are evaluated, this early endpoint would limit the assessment of the therapeutic since it cannot be administered through symptom resolution. It will be necessary for other research laboratories to run their own attenuated protocols to assess whether five-, four-, or potentially even three-day DSS administration regimens are most appropriate for their specific mouse colonies and their specific batch of DSS, which can vary in potency.

Although not a direct measure of inflammation, recording spleen weight at euthanasia is a quick way to correlate inflammation to DSS administration (*Chassaing et al., 2014*). As colitis progresses in the animal, the resulting immune response leads to an increase of immune cells and thus an increase in spleen weight. The increased weight of the spleens from the four-day DSS group is indicative of the robust immune response present despite abbreviating the colitis induction period. Conversely, the decrease in spleen weight observed in the seven-, six-, and five-day DSS groups is likely the result of the severe dehydration and weight loss experienced by these animals.

Shortening of the colon is a hallmark of chemically induced colitis and an often-used metric to compare IBD treatments to controls (*Chassaing et al., 2014*). All groups from this

work demonstrated significant colonic shortening, regardless of administration duration. These data indicate that, despite inducing less severe colitis by most metrics, the four-day DSS model retains measurable colitis symptoms against which to compare therapeutics.

Histological scoring also confirmed that the four-day DSS model produced measurable symptoms of colitis. All DSS groups showed epithelial damage and inflammation compared to the control group, with severity corresponding to administration duration. Histological scores indicate that shorter DSS administration duration generally results in less severe damage to the intestinal epithelium.

Although the etiology of clinical IBD remains unclear, it is generally accepted that genetic predisposition is not the only contributing factor. Diet and lifestyle of western societies are inferred to have some responsibility for the increased prevalence of IBD in countries such as the United States (*Loftus, 2004*; *Kaplan, 2015*; *Colombel & Mahadevan, 2017*). The DSS colitis model is similar in that environmental factors, including diet and animal facility conditions, impact the induced colitis (*Eichele & Kharbanda, 2017*). This makes comparing results between different studies difficult and leads to increased variability of the model, but measures can be taken to ensure that experiments are reproducible with steady colitis induction. All animals in the experiment should be properly acclimated to the animal housing facility before colitis induction begins. Because DSS must be administered via drinking water, the acclimation period should include training animals to drink from the water bottle, rather than the system-wide fresh water supply. Failing to properly acclimate animals results in altered disease model symptoms, which confounds weight loss due to colitis, stress, and/or dehydration.

This study is limited to measurements of weight, fecal scores, intestinal permeability, colon length, spleen weight, and histological evaluation. IBD is a complex disease, and it is likely important to consider other factors, such as tight junction protein expression, the microbiota, and cytokine production (*Martini et al., 2017*). The data presented here suggest that this attenuated four-day DSS model produces results that better recapitulate clinical increases in intestinal permeability. A more robust analysis should be done to confirm that other markers in this model sufficiently represent the clinical disease state.

## CONCLUSION

The DSS colitis model is a powerful tool for studying disease mechanisms and potential therapeutics for IBD. Administering DSS for seven days results in the rapid onset of severe colitis that precludes the testing of epithelium-focused treatments. Here, we demonstrate that an attenuated four-day model induces increases in intestinal permeability that better mimic those of IBD patients while retaining the hallmark signs of chemically-induced colitis. Because this model does not destroy the intestinal epithelium, it will enable the screening of therapeutics that aim to restore or improve barrier function in colitis patients.

## ACKNOWLEDGEMENTS

The authors also thank M Oudhoff and J Gleeson for their helpful discussions and input. Any opinions, findings, and conclusions or recommendations expressed in this material

are those of the author(s) and do not necessarily reflect the views of the National Science Foundation.

### Funding

Funding was provided by the Disruptive Health and Technology Institute at Carnegie Mellon University. Nicholas Lamson received fellowship support from the National Science Foundation Graduate Research Fellowship Program under Grant No. DGE1252522. The funders had no role in study design, data collection and analysis, decision to publish, or preparation of the manuscript.

### Grant Disclosures

The following grant information was disclosed by the authors:
Disruptive Health and Technology Institute at Carnegie Mellon University.
National Science Foundation Graduate Research Fellowship Program: DGE1252522.

### Competing Interests

The authors declare there are no competing interests.

### Author Contributions

- Kyle E. Cochran and Nicholas G. Lamson conceived and designed the experiments, performed the experiments, analyzed the data, prepared figures and/or tables, authored or reviewed drafts of the paper, and approved the final draft.
- Kathryn A. Whitehead conceived and designed the experiments, authored or reviewed drafts of the paper, and approved the final draft.

### Animal Ethics

The following information was supplied relating to ethical approvals (i.e., approving body and any reference numbers):

Animal protocols were approved by the Institutional Animal Care and Use Committee at Carnegie Mellon University (Pittsburgh, PA), and all experiments were conducted in accordance with approved protocols (AR201900009).

### Data Availability

The raw data is available in the Supplemental Files.

### Supplemental Information

Supplemental information for this article can be found online at http://dx.doi.org/10.7717/peerj.8681#supplemental-information.

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
