# Peer review of "Expanding the utility of the dextran sulfate sodium (DSS) mouse model to induce a clinically relevant loss of intestinal barrier function"

_PeerJ, doi:10.7717/peerj.8681_

## Round 0.1 · original submission · Minor Revisions

Please address critiques of the reviewers and amend your manuscript accordingly.

Reviewer 1 ·

Basic reporting

This manuscript reports on a topic that has been extensively investigated over the years.

Experimental design

The experimental design is flawed by the fact that on day 14 only two animals were available for assessing intestinal permeability

Validity of the findings

All data provided.

Additional comments

Limit of the study not described clearly

Reviewer 2 ·

Basic reporting

no comment

Experimental design

On this manuscript, the authors reported that at the experimental colitis model, four-day DSS administration represents the best option by comparing to conventional method that administrate DSS for seven days. They also have highlighted that the four-day protocol mimic better the IBD patients. Therefore, this protocol will be more indicated to investigate possible treatments aiming to restore or improve barrier function in colitis patients

Please to replace “sacrificed” to the scientific term more suitable such as “euthanized”
It will be interesting the authors highlight some aspects involving DSS administration and the short chain fatty acid production. An important aspect to evaluate the intestinal barrier is the microbiota analysis. Probably the concentration of these acids could explain the attenuated injury on intestinal epithelium after 04 days of DSS administration in regarding to others investigated times.
Line 169- The authors should to replace the term “intestinal barrier function” by only intestinal permeability, once the intestinal barrier involves others aspects that were not investigated in this manuscript.

Validity of the findings

Figure 2- Intestinal Permeability
It isn’t clear the relation between the asterisk number (*, **, ****) and the statistical difference level (p=0,005, p<0,0001). In this sense, it would be interesting the authors to clarify better in the Figure 2. For example in the lines 179-180 the authors described “the seven-days DSS group showed the most intense increase in permeability, increasing 35-fold compared to the control” However, in the Figure 2 this results is represented only by one asterisk (p= 0,005), while the six and five days groups were represented by **** p <0,0001.



Conclusion
Although, the four-day DSS protocol has showed similar results of intestinal permeability that could correspond to IBD patients, the outcomes obtained in this manuscript are not enough to suggest this protocol as model to investigate possible treatments to colitis. The literature data have shown that others parameters such as tight junctions proteins, collagen deposition, cytokines, mucus and immunoglobulins production are necessary to understand the complex mechanisms involved in the intestinal barrier. Therefore, the authors should improve the conclusion focusing only the data obtained.

Additional comments

Although, the four-day DSS protocol has showed similar results of intestinal permeability that could correspond to IBD patients, the outcomes obtained in this manuscript are not enough to suggest this protocol as model to investigate possible treatments to colitis. In this sense, others parameters such as immunologic response, citokines and tight junctions proteins should be investigated to support the conclusion due to complexity of intestinal barrier function.

Reviewer 3 ·

Basic reporting

The dextran sulfate sodium (DSS) mouse model of colitis was used to study inflammatory bowel diseases (IBD), which affects more than one million people in the United States. Kyle Cochran et al. optimized the duration of DDS administration to mice to obtain an attenuated DSS colitis model that mimics the conditions of the IBD patients better than any existing DDS mouse model.

The structure of the manuscript conforms to the PeerJ standard. The manuscript is well-written with the proper context of the research questions supported by relevant literature. The figures are of high quality, well-labeled, and described. The raw data were provided.

Experimental design

The research questions are well-defined and appropriate. The methods contain sufficient details to replicate the results.

Validity of the findings

After a systematic investigation, the authors successfully established that the reduced duration of DDS administration to mice better emulate the physiological conditions of IBD patients. The conclusions are well described, related to the original research questions, supported by the results.

Additional comments

Please provide the ethical approval statement for vertebrate animal usage.

---

## Round 0.2 · accepted · Accept

Since all critical issues were adequately addressed and the manuscript was amended accordingly, I am please to accept it for publication in PeerJ.

Reviewer 2 ·

Basic reporting

no comment

Experimental design

The explanation regarding to Figure 2 was enough to clarify the data obtained related to intestinal permeability

Validity of the findings

In the Discussion (lines 318-324) the authors highlighted other aspects related to complexity of intestinal barrier function as mentioned by this reviewer

Reviewer 3 ·

Basic reporting

The authors stated in the Methods section that all the studies were performed following local, institutional, and federal guidelines for vertebrate research.

Experimental design

No comment

Validity of the findings

No comment